# What is multidisciplinary cancer care like in practice? a protocol for a mixed-method study to characterise ambulatory oncology services in the Australian public sector

Bróna Nic Giolla Easpaig [ID],[1] Gaston Arnolda,[1] Yvonne Tran,[1] Mia Bierbaum [ID],[1] Klay Lamprell,[1] Geoffrey P Delaney,[2,3] Winston Liauw,[4,5] Renuka Chittajallu,[1] Teresa Winata,[1] Robyn L Ward,[6,7] David C Currow,[8,9] Ian Olver,[10] Jonathan Karnon,[8] Johanna Westbrook,[1] Jeffrey Braithwaite [ID] [1]

For numbered affiliations see end of article.

**Correspondence to**
Dr Bróna Nic Giolla Easpaig;
brona.nicgiollaeaspaig@mq.edu.au

## ABSTRACT

**Introduction** An understanding of the real-world provision of oncology outpatient services can help maintain service quality in the face of escalating demand and tight budgets, by informing the design of interventions that improve the effectiveness or efficiency of provision. The aims of this study are threefold. First, to develop an understanding of cancer services in outpatient clinics by characterising the organisation and practice of multidisciplinary care (MDC). Second, to explore the key areas of: (a) clinical decision-making and (b) engagement with patients' supportive needs. Third, to identify barriers to, and facilitators of, the delivery of quality care in these settings.

**Methods and analysis** A suite of mixed-methods studies will be implemented at six hospitals providing cancer outpatient clinics, with a staged roll-out. In Stage One, we will examine policies, use unstructured observations and undertake interviews with key health professionals to characterise the organisation and delivery of MDC. In Stage Two, observations of practice will continue, to deepen our understanding, and to inform two focused studies. The first will explore decision-making practices and the second will examine how staff engage with patients' needs; both studies involve interviews, to complement observation. As part of the study of supportive care, we will examine the implications of an introduction of patient-reported measures (PRMs) into care, adding surveys to interviews before and after PRMs roll-out. Data analysis will account for site-specific and cross-site issues using an adapted Qualitative Rapid Appraisal, Rigorous Analysis approach. Quantitative data from clinician surveys will be statistically analysed and triangulated with the related qualitative study findings.

**Ethics and dissemination** Ethical approval was granted by South Eastern Sydney Local Health District Human Research Ethics Committee (no. 18/207). Findings will be shared with participating hospitals and widely disseminated through publications and presentations.

### Strengths and limitations of this study

► Multiple data types and sources are integrated to characterise multidisciplinary care in different settings.
► A two-stage research design facilitates better targeting of observations and interviews in the second stage.
► A strategy of intensive data collection using multiple methods is used to ensure that site-specific data is interpreted within its local context.
► Findings will be based on a rigorous and extensive observational characterisation of Australian oncology outpatient services.

## INTRODUCTION

In the developed world, invasive cancer is the leading cause of death.[1] Globally, the incidence of cancer is increasing[2]; in Australia, cancer incidence is projected to reach 150 000 new cases per year by 2020.[3] Survival rates are improving, with over 400 000 Australians living 5 years post cancer diagnosis.[3] Cancer is understood as a condition requiring medical and supportive care services from diagnosis through long-term survivorship.[4 5] Multidisciplinary care (MDC) is accepted as a best practice model of care provision for oncology services in Australia, and in advanced health systems globally.[6–9] MDC seeks to promote equitable, evidence-based care by a team that combines relevant expertise and enables patients to be involved in decision-making concerning their care.[5 10]

Work has been undertaken to translate MDC principles into practice; but significant challenges remain.[10–12] In this regard, substantial evidence points to gaps between optimal

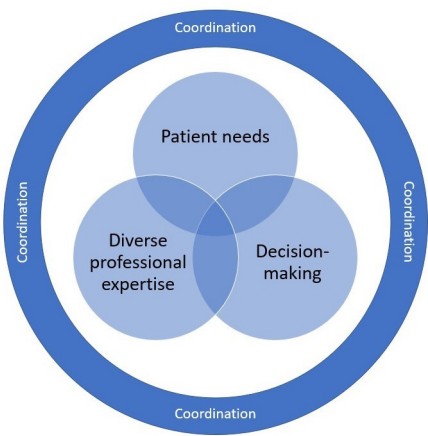

**Figure 1** Key elements of MDC. Source: Authors' conceptualisation of MDC. MDC, multidisciplinary care.

evidence-based care and current care.[13–17] A review of research literature, national health sector reports and local cancer plans identified three key interconnected elements of MDC: diverse professional expertise, clinical decision-making and addressing patients' supportive care needs. In figure 1 the coordination of care is positioned to reflect the way in which it binds and encompasses these elements.

### Patient needs
In addition to medical anti-cancer treatment with curative intent, people with cancer also need medical (eg, pain relief) and non-medical supportive care (eg, information needs). The challenges of addressing supportive needs are an ongoing concern in oncology.[18–21] Meeting these needs requires identification, prioritisation, timely access to relevant professional expertise and treatment planning (eg, in consultation with a psychologist).[18–21] In general, identification of support needs in cancer care has not been systematic. Rather, routine care has relied on the clinical acumen of individual clinicians who identify needs during consultations and make appropriate referrals. Language spoken and level of health literacy also remain challenges in order to 'pitch' information at the level most suitable for each patient. Routinely collecting information from patients, through patient reported measures (PRMs), has been promoted as a mechanism to identify unmet supportive needs.[22–24] The Edmonton Symptom Assessment Scale,[25] Distress Thermometer[26] are PRMs which some of the outpatient clinics (OPCs) will be introducing into routine cancer care during the period of study, and are expected to change the management of patients' supportive needs.[27–29] This study focuses on the provider perspective; the patient's support networks are also important for addressing supportive needs, but these are beyond the study's scope.

### Diverse professional expertise
MDC is overseen by a core team which, while varying by cancer type, usually consists of pathologists, radiologists, medical oncologists, radiation oncologists, surgeons,

nurses and allied health professionals.[5 10 30] Other specialist expertise can also be part of the core team, or can be added as required. This may include expertise such as in palliative care or pre-rehabilitation and post-rehabilitation over the course of the patient's journey.[5] This approach is intended to use cross-disciplinary expertise to the fullest benefit of the patient and offer shared decision-making mechanisms for clinicians.[10 31] To develop an understanding of MDC we therefore need to harness the views of a diverse range of professionals and observe interprofessional collaboration in practice.

### Decision-making
Multidisciplinary team meetings (MDTMs) are a fulcrum for the clinical coordination of MDC.[11 12] Research suggests that MDTM discussions influence diagnosis, staging and treatment planning.[32–34] It is important to understand the organisation of MDTMs and the resources required to support them. Studies reveal variability in the organisation of MDTMs, who is discussed and when, and the ways that clinicians engage to support clinical decision-making.[9] While MDTMs play a central role, clinicians make decisions outside MDTMs, without any consultation during routine care provision, or in consultation with other clinicians who may or may not be part of the team. In addition, decisions can be made at the meetings but evidence suggests that the recommendations are not always followed and in many meetings there can be disagreements in opinion.[35] How these issues are managed with the patient are likely to be highly individualised. We also seek to understand this behaviour.

### Coordinated care
Among the top priorities for coordination are: (1) ongoing screening of patients' supportive needs and appropriate care in response and (2) the timely and appropriate delivery of care across the patient's journey.[36] Public hospital OPCs are tasked with providing care for diverse types of cancer across the patient journey, from diagnosis, through treatment and into survivorship. The models of coordination employed may be non-uniform; hybrid approaches may be used within a setting (eg, simultaneously offering tumour stream-specialised care coordination, with generalised care coordinators covering all others).[37 38] Sub-optimal coordination can lead to fragmented care, conflicting information provided to patients, impaired access to appropriate services and inefficient use of the time and energy of health professionals and patients.[21 39 40]

### Research framework
The investigation described in this paper is the first major phase of field research undertaken by the newly established National Health and Medical Research Council Centre for Research Excellence in Implementation Science in Oncology (CRE-ISO), administered by the Australian Institute for Health Innovation at Macquarie University. In healthcare, implementation science focuses

on understanding processes and methods for supporting the uptake and integration of effective interventions to improve healthcare practices and outcomes.[41] The research described here will characterise MDC provision in two Sydney metropolitan Local Health Districts (LHDs), government agencies responsible for managing and providing public health services within a specified geography, usually of approximately one million residents in order to provide a foundational understanding of the realities of oncology service provision. This understanding will inform subsequent phases of research undertaken by the Centre and will guide the selection of implementation approaches, some of which will focus on developing and testing tailored interventions to address identified evidence-practice gaps.

Critical to this study is engagement with the context of service provision and its complexity; of the 'interacting and interdependent' components of a health service that comprise MDC provision in hospitals.[42] Multiple sources of data will be harvested and integrated, exploring the formalised processes and local practices undertaken within the interconnected networks of semi-autonomous professionals that together shape care provision.[43] Systems of care provision in these settings are rarely static, and opportunities will be taken to examine the uptake of new practices, such as the introduction of PRMs, to learn about the processes that inhibit or promote effective adoption. This research will also seek to identify generic barriers and facilitators that impede or promote best practice, as background information to assist with the design of future interventions.

### Research aims (RA)

The study aims to:

RA1. Develop understanding of cancer services in OPCs by characterising the organisation and practice of MDC;

RA2. Explore, in-depth, selected key elements of MDC:

i. Clinical decision-making in MDC and how doctors engage with decision-making support mechanisms in practice; and

ii. How health professionals engage with patients' supportive needs before and immediately after the introduction of PRMs.

RA3. Identify general barriers to, and facilitators of, the provision of MDC.

A mixed-methods research design will be adopted: largely comprised of a variety of qualitative methods, complemented by a single quantitative survey. Methods adapted from ethnography will be harnessed, to facilitate intensive and efficient data collection; rapid ethnography will be used to reduce the burden of research on organisations and participants while maintaining rigour and richness.[44 45] Careful thought has been given to the demands of multisite research and we adopt recommended strategies used to enhance validity and maintain capacity to adequately engage with the complexities of the settings.[46]

The methodological principles embedded in the design are: exploring everyday practices and the organisational relations that shape them, by observing participants at work in OPC settings; harnessing a plurality of participant perspectives, by interviewing professionals about their work and generating rich descriptive accounts of MDC contexts by using analytical and interpretative approaches formulated for multisite data.[47] A critical realist philosophical position is taken, which is comprised of a realist ontological assumption and a critical epistemology.[48 49] From this position it is assumed that a given account may not perfectly reflect reality; rather, accounts may be contextually and structurally mediated, offering partial insight into complex phenomena.[50 51]

### Design

A staged design will be used. Prior to study commencement, profiles and structural descriptions of each hospital setting were developed, using publicly-available information. In *Stage One,* an overview of the organisation of practice and models of care will be derived through interviews with 'Navigators' (eg, OPC staff with an overview of care pathways such as nurse unit managers or care coordinators), review of unpublished policies and procedures and unstructured observations of care provision. In *Stage Two*, we enrich the accounts of these settings through additional observation in OPCs. Together, these methods address the first research aim (RA1) and provide a foundation for the second (RA2) and third (RA3).

*Stage Two* also includes focused studies, undertaken in parallel. In Study 2.1, interviews with doctors and observations of cancer MDTMs provide insight into clinical decision-making practices in MDC and how doctors engage with decision-making support mechanisms (RA2.i). Engagement with patients' supportive needs will be explored in Study 2.2 (RA2.ii), using a two part, mixed-method design consisting of interviews with health professionals, observation of cancer MDTMs and a 'clinician readiness' survey; all research elements will be performed before introduction of new PRMs, and repeated after they have been introduced.

Barriers and facilitators to the provision of MDC (RA3) will be elicited as part of the interviews in each study and enhanced by our observational studies in *Stages One* and *Two*. The study design is informed by the Consolidated Criteria for Reporting Qualitative Research.[52] The study will run from the 28th of February, 2019, up to the 31st of December, 2020.

### Selection of sites and participants

The investigation will be undertaken within two metropolitan LHDs, each containing three participating public hospitals with OPCs. Collectively, the OPCs offer an extensive range of consultation, coordination, wellness and outpatient treatment services. Each LHD contains two large hospitals which provide an extensive range of services and a smaller hospital offering a more limited range of services, closely linked with larger centres (eg, including the provision of teleconferencing for meetings).

*Sites and meetings:* In *Stage One* all sites will be studied, to understand each hospital setting. In *Stage Two*, maximum variation sampling[53] will be used to allocate the 'continuing observation' sessions to participating sites. As there are differences in the services between sites (eg, some hospitals offer a smaller range), observation sessions will be allocated to best capture the diverse range of practices and processes useful in the characterisation of MDC within and across sites. Purposive sampling will be used to select the MDTMs approached to participate. We will generate a sample comprising a range of tumour streams including common and rarer cancers. Study 2.1 will largely be conducted in one LHD, with study 2.2 in the other.

*Participants:* The participant inclusion criteria for interviews, surveys and health professional observations is generally broad: core medical staff (eg, medical oncologists, surgeons, radiation oncologists), nursing staff (eg, clinical nurse consultants, cancer care coordinators) and allied health staff (eg, social workers, psychologists). However, participation in *Stage One* 'Navigator interviews' will be restricted to staff with an overview of care pathways (eg, cancer care coordinators, nurse unit managers, managers of clinical streams). Similarly, in Study 2.2 the 'clinician decision-making' interviews will be undertaken with specialist doctors (eg, medical oncologists, surgeons, radiation oncologists) working in cancer services.

*Recruitment:* Information about the research including its aims, an introduction to the researchers undertaking data collection, contact details and participant information sheets will be disseminated through email, flyers and posters, and through information sessions for target professional groups at each site. Snowball sampling will also be used; participants will be encouraged to discuss the project with colleagues. Letters will be sent to the Chairpersons of cancer MDTMs and to the authorities responsible for the OPCs, to provide information about the research and request permission for the observations. These letters will include 'Health Professional Information Sheets' which recipients will be encouraged to share with relevant colleagues as a component of pre-approval consultation.

*Sample size ranges:* The Study design features, presented in table 1, summarise the stages of the research, the data collected in each stage and the target number of observations, interviews and surveys. The sample sizes designated for the interview components are guided by principles identified in reported research literature, as well as Malterud, Siersma and Guassora's (2016) model for generating 'information power' in qualitative studies. The model accounts for the following factors when estimating a sample range: the study aim, specificity of the sample, the envisioned quality of the interview interaction, analysis strategy and established theory.[54] The number of proposed interviews in Study 2.2. is also guided by the sample size needed for the clinician readiness survey that is undertaken as part of the interview. The sample size allocated for the individual health professional observations reflects the narrow and specific focus of gaining a deep understanding of the roles these professionals play in MDC. The number of sessions proposed for the observations of the OPCs and MDTMs are comparable with similar qualitative studies[55] and are appropriate for multisite ethnographic studies.[56]

*Research Team:* The investigation will be overseen by a team with extensive experience in health services and

| Phase/Stage | Text | Interviews | Survey | MDTM Observations | UndirectedObservation Sessions* |
|---|---|---|---|---|---|
| **Table 1** Study design features | | | | | |
| **Stage One** | | | | | |
| **Stage OneMulti-site characterisation** | Review of policies and procedures relevant to interview discussion | Navigator (n=8-16) Decision-making (n=3-8) Supportive needs (n=4-10) | n/a | Cancer MDTMs (n=3–8) | Initial: OPCs (n=15-25) Individual (n=3-6) |
| **Stage Two** | | | | | |
| **Study 2.1 Decision-making** | Review of policies and procedures relevant to interview discussion | Decision-making (n=15–30) | n/a | Cancer MDTMs (n=3–8) | Ongoing: OPCs (n=15) Individual (n=7) |
| **Study 2.2. Supportive needs** | | Baseline (n=15-30) Follow-up (n=15-30) | Baseline (n=30-50) Follow-up (n=30-50) | Cancer MDTMs (n=3–8) | |
| **Estimated totals** | n/a | 60–124 | 30–50 | 9–24 | OPCs (n=30-40) Individual (n=10) |

Table key

| | |
|---|---|
| | Undertaken at all sites |
| | Predominantly undertaken in LHD 1 |
| | Predominantly undertaken in LHD 2 |

*1 observation session= half day or 3 to 4 hrs.

LHD, Local Health District; MDTM, multidisciplinary team meeting; OPCs, outpatient clinics.

quality improvement research, including leaders in implementation science and oncology. Fieldwork will be guided by principal investigators with direct clinical and research experience in oncology service provision, in leadership roles in the LHDs. Data will be collected by GA (Senior Research Fellow, PhD, male), BNGE (Research Fellow, PhD, female) and TW (Research Associate, Masters, female) who are trained in qualitative, quantitative and mixed-methods investigation. Each of these researchers has fieldwork experience across a range of health services. Qualitative analysis will be conducted by BNGE and TW, while YT (Research Fellow, PhD, female) will analyse quantitative data. Appropriately qualified health professionals who join the research team for *Stage Two* will be trained and supervised.

## Data collection

As part of protocol development, reviews of the literature were conducted to conceptualise and identify key issues in MDC provision and to scope the methodological scholarship relevant to the design. Publicly-available annual reports and cancer plans from each health district were analysed to generate a description of cancer services at each hospital. These profiles were used to map key areas for data collection. Implementation science approaches emphasise the importance of the local context of adoption and engagement with its stakeholders.[41–43] During this stage, input was sought from oncology management to ensure the local relevance of the research and the feasibility of the proposed methods.

## Stage one - multisite characterisation study

*Interviews:* Semi-structured 'Navigator' interviews will be undertaken with professionals who have overview roles (eg, cancer care coordinator), oriented by care pathway maps, to establish the models of care and processes involved in the patient journey (n=8 to 16).[57 58] The care pathway maps were derived from the stages of MDC depicted by Fennell, Das, Clauser, Petrelli and Salner,[59] focused on the diagnosis and OPC treatment stages (figure 2). Through the interviews, maps will be reworked to reflect the Australian services context, and the differing trajectories followed by different: (a) tumour streams (accounting for tumour-specific services) and (b) risk categories and complexity. Where interviewees refer to specific policies and procedures, documentation will be formally requested to help situate the discussion within its policy context. Interviews are expected to take approximately 40 min and will be audio-recorded and transcribed.

*Observations:* Three types of observations will be made in the research:

▶ Unstructured observations of OPCs will provide an understanding of the real-world, routine organisation and actual delivery of MDC in these settings (eg, we will observe ad hoc and planned collaboration between professionals from differing disciplines during the care delivery process; n=15 to 25).

▶ Observations of individual health professionals will capture the key activities undertaken, to understand how their work contributes to the provision of MDC (n=3 to 6 individual health professionals observed for half or all of their day).

▶ Observations of MDTMs will be conducted using a semi-structured template (figure 3) informed by previous MDTM observational research to guide what we observe and record, and systematise our description of meetings (n=3 to 8 meetings).[11 60] Field notes will record observations of: professional groups in attendance, specific roles, support provided to administer the meeting (ie, technology), organisation of cases, clinical decision-making, patient-centred care and organisation of follow-up.

All observations will be recorded in handwritten field notes which will be used to generate rich accounts of care.

*Transition to Stage Two:* During *Stage One*, we will pilot elements of the clinician interviews about decision-making (n=3 to 8) and patients' supportive needs (n=4 to 10) to be undertaken in *Stage Two*, and interview prompts will be refined. Where eligible health professionals indicate interest in joining the research team for *Stage Two* fieldwork, they will be briefed and provided with necessary training and support.

## Stage two - MDC studies

This stage will enrich accounts of MDC provision via 'ongoing characterisation' (ie, n=3 to 6 further observations of individual health and n=15 unstructured observations of OPCs), at the same time contextualising the focused studies detailed below.

### Study 2.1 - clinical decision-making

*Interviews:* Doctors will participate in semi-structured interviews that explore their decision-making practices and their engagement with support mechanisms (n=15 to 30). Topics will include their perceptions of decision-making mechanisms, the decision-making support available to them, their own decision-making practices and their use of MDTMs at key points in the patient journey. A conceptual map of the patient journey (figure 4) will be used to orient the interview, with questions tailored to each stage. Barriers to and drivers of efficient clinical decision-making will be explored. Interviews are expected to take approximately 40 min and will be audio-recorded and transcribed.

*Observation of MDTMs*: Meetings will be observed using a semi-structured template (figure 3), focussing on the 'clinical decision-making' component (n=3–8). Observational data will capture information about this decision-making mechanism and contextualise interviews.

### Study 2.2 Patients' supportive needs

*Interviews:* Prior to the introduction of PRMs, semi-structured interviews will be conducted with health professionals about how they engage with patients' supportive needs in their current practice (n=15 to 30). Topics will

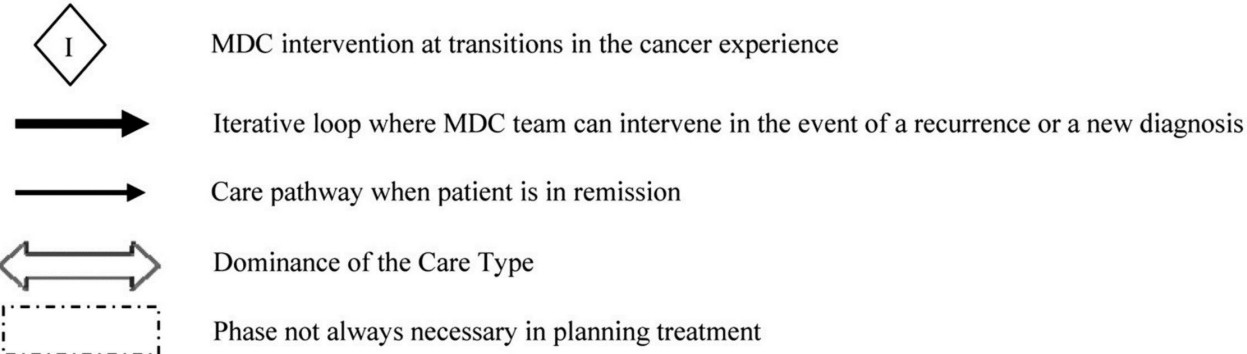

**Figure 2** MDC interventions at transitions in cancer care. Source: Fennell et al[59] (adapted from Zapka et al).[65] MDC, multidisciplinary care.

include: models of care, their individual role, identification and screening, management and any barriers and facilitators experienced in identifying and responding to supportive needs. The interviews will also explore preparations for and expectations of PRMs.

Following the adoption of PRMs, health professionals will again participate in semi-structured interviews (n=15 to 30). These will focus on the adoption and implications of PRMs in relation to the topics addressed in pre-introduction interviews. Pre-introduction and post-introduction interviews are expected to take approximately 40 min and will be audio-recorded and transcribed.

*Surveys:* An eight-item Clinician Readiness Survey[61] will be completed by health professionals before and after the adoption of PRMs (n=30 to 50 in each phase). The survey will assess the current level of preparedness, and changes over time. This survey will take approximately 5 min to complete and will be performed as part of the interviews and will also be distributed more widely to achieve the desired sample size.

| MDTM & Hospital: | Notes |
|---|---|
| Attendance of health professional groups represented at MDT meetings by various health professional groups | |
| Chairperson and any meeting specific roles allocated | |
| Technological, equipment and administrative support in MDT meetings | |
| Organisation of the presentation of cases including<br>• The order<br>• Any prioritisation strategies<br>• Any patient inclusion criteria for meetings | |
| Clinical decision-making<br>• Discipline of professional input<br>• Types of information considered<br>• Evidence and/ or research incorporated into meetings | |
| Patient-centred care & Supportive care needs | |
| Organisation of any follow up tasks | |

**Figure 3** MDTM data collection template. Source: Developed by authors based on Harris *et al*[60] and Rankin *et al*.[11] MDC, multidisciplinary care; MDTM, multidisciplinary team meetings.

*Observation of MDTMs*: Meetings will be observed using a semi-structured template (figure 3), focussing on the 'patient-centred care and supportive care needs' component (n=3 to 8). Data will capture information about engagement with supportive needs in MDTMs and contextualise the interviews.

## Data analysis

*Interview data:* Basic descriptive data collected during interviews (eg, listing of team members involved in different points of the patient journey) will be collated across interviews then categorised and organised within a matrix. The discursive data will be coded by two researchers and thematically analysed, oriented by an inductive approach.[62] Where process maps are used (eg, navigator and decision-making interviews), these will frame the analysis. Constant comparative analysis will be applied to interpret data gathered before and after the introduction of PRMs. This strategy accounts for the cross-site, multiple-source nature of the data. Our approach for making sense of cross-site data is informed by the 'Qualitative Rapid Appraisal, Rigorous Analysis' methodology developed by Phillips *et al*.[45] First, all 'like' data (eg, all interview data) is analysed at a site, followed by intrasite analysis of all data from one OPC. Intersite analysis is then employed to consider the data across the sites.[45]

*Observational data:* Field notes will be indexed to identify the specific setting/MDTM meeting, key processes and events and study focus. The notes will be integrated within hospital cancer service profiles to elaborate descriptions of OPC settings (ie, the organisational features of care). Field notes from MDTM and OPC observations will then be developed into accounts that contextualise data collected in the focused studies. Data collected from individual health professional observations will be developed into an account of the role.

*Clinician readiness survey data:* Analysis will include a repeated-measures analysis of variance to test for differences in clinician attitudes before and after PRMs roll-out. If possible, subgroup analyses will test for differences in attitudes between different demographic and clinical subgroups. The quantitative data will be used to complement the interview data on this topic. Based on known effect sizes from results of the Willis *et al*[61] paper using the Outcome Measurement Questionnaire to assess change in response to training, using a power of 0.8 and alpha of 0.05, the minimum sample size was estimated as n=32 participants (ie, to detect a change in mean scale score from 4.09 pre-introduction to 4.49 post-introduction, with a SD of 0.57).

## Measurement of outcomes

The main outcomes of the investigation are aligned with the aims. Through this investigation we will generate a rich and comprehensive description of the organisation and practice of MDC. We will document practices of clinical decision-making including the role of the MDT in such practice. We will produce an account of how patient supportive needs are engaged with and the implications that the introduction of PRMs has for this engagement. The factors which impede or promote practice will be identified and used to inform subsequent CRE-ISO activity.

For the qualitatively-driven research, assessment of the outcomes involves judgement of the adequacy with which the research aims were addressed. This is largely reliant on the quality of the data collected and the rigour of its analysis. Sufficiently detailed data are needed from a range of sources to produce rich descriptions of MDC practice, including its diversity. Fieldwork and analytical strategies should allow findings to be situated within OPC

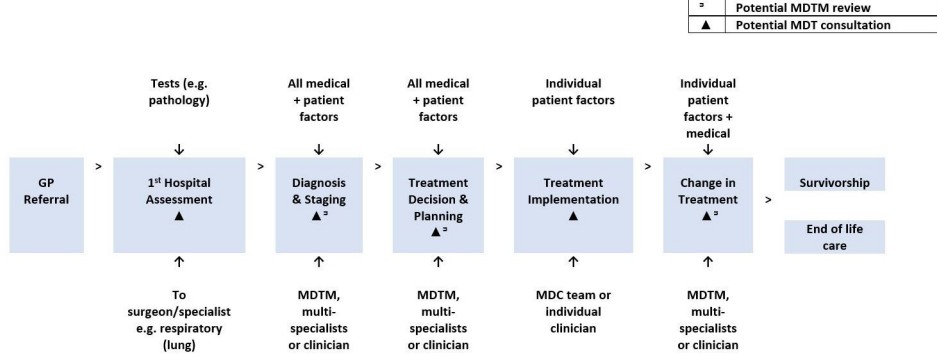

**Figure 4** Mapping of points of diagnosis and treatment decision-making in MDC. Source: Authors' adaption of Fennell *et al*.[59] GP, general practitioner; MDC, multidisciplinary care; MDTM, multidisciplinary team meetings.

contexts as well as revealing points of connection and contradiction between sites. Characterisation of MDC provision will be achieved through credible, adequately complex and well-grounded accounts. Multi-method data collection (including a quantitative component) allows for consideration of various sources of input, and triangulation; a team-based approach to analysis will facilitate ongoing reflection and monitoring.

## ETHICS AND DISSEMINATION

Informed written consent will be obtained from interviewees and from health professionals who are individually observed. Interview and observation arrangements will be made with health professionals at least 1 week in advance to ensure they have time to prepare. Participants may withdraw, without justification, up to 1 week post interview/observation or whenever the data is de-identified.

Posters will be displayed, and information sheets will be available during observational sessions so that those present can raise concerns or decline participation. This can be done at any time up until the observational data is collected. The research team will not collect any information about patients and, guided by health professionals, will minimise exposure to direct patient interactions. Researchers will answer questions posed about their presence and provide information sheets that contain ethics committee details. In the event of participants experiencing discomfort or distress arising from the research, researchers will facilitate access to appropriate support services.

*Data storage and protection:* Data will be stored on a password-protected computer and on a secure university server, accessible only to the research team. Data will be de-identified and separated from participant identifiers. Research records will be retained for at least 5 years post-study completion or last publication, in accordance with Section 2.1.1 of the Australian Code for the Conduct of Responsible Research (2007)[63] before secure destruction.

*Dissemination:* Findings will be disseminated to academics, professionals and the public through publications and presentations. Tailored reports will be provided to participating hospitals and presentations given to participating professional groups.

## Patient and public involvement statement

The coordinating research institution routinely consults with patients, their representatives and the general public, to ensure that adequate input is secured for research programme; a key partner is the Consumers Health Forum of Australia. This specific project was developed without detailed patient involvement; patients were not invited to comment on the study design or to the writing or editing of this protocol. This project will help us to describe service structure and organisation, and to learn from the perspectives of MDC providers. The experiences of patients are vital to understanding

these services, including their views on the barriers and facilitators to high-quality care. Patient experiences will be engaged with dedicated, patient-centred studies within the CRE-ISO programme of research. For example, in preparation, we have already looked at data from UK Patient Experience Surveys,[64] and are in the process of accessing Australian data from a similar source.[64]

## Impact & Significance

The research seeks to capture a diverse range of practices and processes useful in the characterisation of MDC, with attention to site-specific nuances that may be relevant to future improvement efforts. The accumulative characterisation will be combined with focused, in-depth studies of clinical decision-making and processes for meeting patients' supportive care needs. The research involves rigorous and extensive observations of Australian public oncology OPCs and MDTMs; it offers a substantial opportunity for the generation of new and critical insights. The findings will serve as a foundation for the design of studies to address identified evidence-practice gaps in cancer care.

**Author affiliations**
[1]Australian Institute of Health Innovation, Macquarie University, Sydney, New South Wales, Australia
[2]Liverpool Cancer Therapy Centre, Liverpool, New South Wales, Australia
[3]University of New South Wales South Western Sydney Clinical School, Liverpool, New South Wales, Australia
[4]Saint George Hospital Saint George Cancer Care Centre, Kogarah, New South Wales, Australia
[5]Saint George Hospital Clinical School, University of New South Wales, Sydney, New South Wales, Australia
[6]Faculty of Medicine and Health, University of Sydney, Sydney, New South Wales, Australia
[7]Prince of Wales Clinical School, University of New South Wales, Sydney, New South Wales, Australia
[8]College of Medicine and Public Health, Flinders University, Adelaide, South Australia, Australia
[9]Faculty of Health, University of Technology Sydney, Sydney, New South Wales, Australia
[10]Division of Health Sciences, University of South Australia, Adelaide, South Australia, Australia

**Contributors** JB, GPD, WL, RLW, DCC, IO, JK and JW led the conceptualisation of the work and the development of the design was led by JB, GPD, WL, GA, BNGE, YT, MB, KL, RC and TW. With guidance from the other authors, BNGE wrote the first draft and all authors contributed to revising key content to develop the final version. All authors have seen and approved of the final manuscript.

**Funding** JB receives funding to support this work from NHMRC grants APP9100002 and APP1135048.

**Competing interests** None declared.

**Patient consent for publication** Not required.

**Ethics approval** Ethical approval was granted by South Eastern Sydney Local Health District Human Research Ethics Committee (no. 18/207).

**Provenance and peer review** Not commissioned; externally peer reviewed.

**Data availability statement** There are no data in this work.

and indication of whether changes were made. See: https://creativecommons.org/licenses/by/4.0/.

**ORCID iDs**

Bróna Nic Giolla Easpaig http://orcid.org/0000-0001-6787-056X

Mia Bierbaum http://orcid.org/0000-0002-7037-4708

Jeffrey Braithwaite http://orcid.org/0000-0003-0296-4957

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
