## [Reviewer comments · BMJ Open]

ARTICLE DETAILS

TITLE (PROVISIONAL)	What is Multidisciplinary Cancer Care Like in Practice? A protocol for a mixed-method study to characterise ambulatory oncology services in the Australian public sector
AUTHORS	Tran, Yvonne; Bierbaum, Mia; Lamprell, Klay; Delaney, Geoffrey; Liauw, Winston; Chittajallu, Renuka; Winata, Teresa; Ward, Robyn; Currow, David; Olver, Ian; Karnon, Jonathan; Westbrook, Johanna; Braithwaite, Jeffrey

VERSION 1 – REVIEW

REVIEWER	Mackenzi Pergolotti ReVital Cancer Rehabilitation, Select Medical Colorado State University USA I receive a salary from Select Medical.
REVIEW RETURNED	17-May-2019

GENERAL COMMENTS	Thank you for inviting me to review this important work. I think this could be a giant step forward in better understanding how MDC can, and could work in helpful and sustainable manner for patients with cancer. I have a few concerns/comments: The most major concern is the decision to not include patients. Please take room to really describe why patients were not included- especially in terms of understanding the facilitators and barriers to care. It appears cancer rehabilitation was excluded from list of MDT providers, yet they are considered a major team member. How are they not included when it comes to "practical" types of intervention when the unmet needs can include both physical and mental health- including the strong desire of survivors to not be a "burden" to others and the importance of maintain independence and being physically active? Also, what about MDC care during treatment (and even prehabilitation- care at diagnosis but before treatment) how does this fit within the model? How does coordination fit within figure 1? Why is it outside, and how does this impact patient needs, and how does it look now- versus the ideal? More clarity regarding how and what will be qualitatively measured in team meetings. Need more clarification on statistical design for quant portion- you may have sample size for changes- but will 30-50 folks allow for sub-group analysis? Will snowball sampling not bias your sample as to folks willing and able to discuss MDC?
---

	In figures, what about patients on maintenance care- say chemo- are they considered palliative, survivors? This needs to be clarified. More data/literature is need to make argument that primary care really does all decision making for survivors after primary treatment completes. Also-palliative care and survivorship care are separate boxes- do they not overlap? what about patients that would like to work on both? Please explain how recent studies describing early and often use of palliative care relates to figures and study. Minor comments: Not sure what you mean by Please define clearly- "time deepen" Page 7- what do you mean by "these trajectories frame cancer..." I think you need more detail about the needs of survivors to frame the argument around supportive and MDC- not just that the numbers are increasing. Page 7- first para- last sentence- please explain patient involvement here Page 8- use patient first language (not "cancer patient") Explain "practical" do you mean functional? what about physical? How does PRM use relate to health literacy- page 8 Citation needed - page 9- first para, second to last sentence ("disagreements in opinion") Coordinated care- wouldn't this be #2 than #1? Don't you need screening to know what needs are before delivery of service? In methods- 30-50 interviews of 40 mins in length, plus observations, seems like a ton of data--why did you choose those numbers? What if you reach saturation before hand? What is the justification behind it?
--	---

REVIEWER	Kristen Mccarter University of Newcastle Australia
REVIEW RETURNED	14-Jun-2019

GENERAL COMMENTS	This protocol describes a suite of studies that will describe clinician attitudes and provision of multidisciplinary oncology care. This work will be an important first step in designing interventions to improve the provision of this care. General comments: - The authors mention implementation science approaches to these studies. Is there a specific framework being followed? Specific comments: - How will the authors observe ad hoc collaboration between professionals? Will the half day observations be sufficient to observe this? - What are the PRMs that are going to be introduced? - Are there any other methods that could be used in combination with observation of clinician practise? Clinicians may be biased in their practice from the knowledge that they are being observed. Further, if no patient interactions are observed, the authors will be relying on clinician self-report? Whilst this may unavoidable, could the data be supported by medical record notes of decision making, electronic referrals etc?
--

	 - Please describe who will conduct the qualitative analysis and their relationship to the research team/authors - BMJ guidelines state that the dates of the study should be included in the manuscript. Please include. - The main outcomes of the studies need to be described
--	--

VERSION 1 – AUTHOR RESPONSE

Reviewer: 1
General comments

1. The most major concern is the decision to not include patients. Please take room to really describe why patients were not included- especially in terms of understanding the facilitators and barriers to care.

We are in complete agreement with the reviewer that the experiences of patients are vital and believe that we will better engage with patient perspectives through dedicated patient-centred studies within the Centre for Research Excellence in Implementation Science in Oncology (CRE-ISO) program of activity. The text in section “Patient and Public Involvement Statement” (p 24) has been elaborated to emphasise this point and to note the importance of learning about the facilitators and barriers patients experience. The text now reads:

This project will help us to describe service structure and organisation, and to learn from the perspectives of MDC providers. The experiences of patients are vital to understanding these services, including their views on the barriers and facilitators to high-quality care. Patient experiences will be engaged with in dedicated, patient-centred studies within the CRE-ISO program of research. For example, in preparation we have already looked at data from UK Patient Experience Surveys[63] and are in the process of accessing Australian data from a similar source.

2. It appears cancer rehabilitation was excluded from list of MDT providers, yet they are considered a major team member. How are they not included when it comes to "practical" types of intervention when the unmet needs can include both physical and mental health- including the strong desire of survivors to not be a "burden" to others and the importance of maintain independence and being physically active?

We thank the reviewer for reminding us of the importance of pre- and rehabilitation in the care process. The text on page 8 has been revised to reflect this:

Other specialist expertise can also be part of the core team, or can be added as required. This may include expertise such as in palliative care or pre- and rehabilitation over the course of the patient’s journey.

3. Also, what about MDC care during treatment (and even prehabilitation- care at diagnosis but before treatment) how does this fit within the model?

We concur that, ideally MDC would be provided across the patient’s journey and we have revised the text on page 8 to make reference to this:

This may include expertise such as in palliative care or pre- and rehabilitation over the course of the patient’s journey.

We have also removed reference to “MDC” at the “treatment implementation” stage in Figure 4, lest we give the impression that this care only applies at this stage. This text has been substituted for “care team”. The text now reads:

Care team or individual clinician

4. How does coordination fit within figure 1? Why is it outside, and how does this impact patient needs, and how does it look now- versus the ideal?

Through reviewing pertinent literature, the challenges for coordinating care are identified and described on page 9. Ideally, coordination would underpin and facilitate each of the other elements depicted in Figure 1. The following revision has been made on page 7 to reflect this:

In Figure 1 the coordination of care is positioned to reflect the way in which it binds and encompasses these elements.

5. More clarity regarding how and what will be qualitatively measured in team meetings.

Measurements will not be made in multidisciplinary team meetings. Instead, we use the semi-structured template shown in Figure 3 to guide and focus our observation of meetings. We have revised the text on page 18 to clarify the descriptive nature of the observation:

Observations of MDTMs will be conducted using a semi-structured template (Figure 3) informed by previous MDTM observational research to guide what we observe and record, and systematise our description of meetings (n=3-8 meetings).

6. Need more clarification on statistical design for quant portion- you may have sample size for changes- but will 30-50 folks allow for sub-group analysis?

We understand that with 30-50 surveys the power for subgroup analyses is reduced, thus the purpose for using this information will be to complement to the qualitative analysis (p 21). Although we may not find statistical significance with reduced power unless differences are stark, to get a better understanding from the surveys we hope to explore other techniques such as effect sizes and equivalence analysis for subgroups. The text on page 21 has been revised to convey this:

If possible, subgroup analyses will test for differences in attitudes between different demographic and clinical subgroups. The quantitative data will be used to complement the interview data on this topic.

7. Will snowball sampling not bias your sample as to folks willing and able to discuss MDC?

This is an important limitation of snowball sampling which will be acknowledged and reflected upon during analysis and in the reporting of the data.

8. In figures, what about patients on maintenance care- say chemo- are they considered palliative, survivors? This needs to be clarified. More data/literature is need to make argument that primary care really does all decision making for survivors after primary treatment completes. Also-palliative care and survivorship care are separate boxes- do they not overlap? what about patients that would like to work on both? Please explain how recent studies describing early and often use of palliative care relates to figures and study.

We are appreciative to have the opportunity to address these issues.

Figure 2 presents Fennell, Das, Clauser, Petrelli and Salner’s (2010) depiction of MDC, in which primary care is positioned as the dominant form of care following treatment. In our work we use the diagram by Fennell et al. (2010) as a basis from which to develop our own tailored maps that reflect

the Australian context of service delivery and the specific care pathways stemming from the interviews in Stage One. On page 17 the text has been revised to emphasise the points made above: Through the interviews, maps will be reworked to reflect the Australian services context, and the differing trajectories followed by different: a) tumour streams (accounting for tumour-specific services); and b) risk categories and complexity.

Figure 4 shows the map we will use in the interviews in Study 2.1 "Clinical Decision-making". We did not intend to indicate that primary care was the principal provider in survivorship, but understand how what we have written could be construed in that way; therefore we have updated the final stage of this map to note the involvement of:

Primary, specialist and other care providers

We recognise that palliative care may be provided at any stage during the process, including at the earlier stages of the patient journey. As such, we have removed palliative care in the final stage of the Figure 4 map, considering it instead as a potential form of treatment. We replaced the term 'palliative care' with the term 'end of life care' in this location.

In this paper we do not focus in depth on any specific form of treatment and believe that the inclusion of recent studies describing early and frequent use of palliative care would not be in alignment with this approach.

Minor comments

9. Please define clearly- "time deepen"

Thank you for raising this issue. "Time-deepen" is the idea that when researchers work with a smaller unit of field time, it is important that strategies are in place which maximise the collection of rich and complex data. This phrase has been excluded on page 11 as the final part of this sentence more accurately captures this idea:

Careful thought has been given to the demands of multi-site research and we adopt recommended strategies used to enhance validity and maintain capacity to adequately engage with the complexities of the settings.

10. Page 7- what do you mean by "these trajectories frame cancer..." I think you need more detail about the needs of survivors to frame the argument around supportive and MDC- not just that the numbers are increasing.

We have revised this sentence on page 7 for clarity. It now reads:

Cancer may be understood as a condition requiring medical and supportive care services from diagnosis through long-term survivorship.

11. Page 7- first para- last sentence- please explain patient involvement here

This sentence has been updated to reflect the involvement of patients as detailed in the Cancer Australia "Principles of Multidisciplinary Care". The text on page 7 now reads: MDC seeks to promote equitable, evidence-based care by a team that combines relevant expertise and enables patients to be involved in decision-making concerning their care.

12. Page 8- use patient first language (not "cancer patient")

On page 7 "cancer patients" has been replaced by: people with cancer

13. Explain "practical" do you mean functional? what about physical?

We appreciate that it might not be clear what is meant by this term. The sentence uses examples to illustrate what would be considered a need of this nature, rather than providing an exhaustive listing of all of these needs. On pages 7 and 8 this sentence has been revised to present one, clear example: In addition to medical treatment, people with cancer also need medical (e.g., pain relief) and non-medical supportive care (e.g., information needs).

14. Citation needed - page 9- first para, second to last sentence ("disagreements in opinion")

On page 9 the following research is cited to support this statement:
Hamilton DW, Heaven B, Thomson RG, et al. Multidisciplinary team decision-making in cancer and the absent patient: a qualitative study. *BMJ Open* 2016;6:e012559. doi:10.1136/bmjopen-2016-01255

15. Coordinated care- wouldn't this be #2 than #1? Don't you need screening to know what needs are before delivery of service?

Thank you for drawing our attention to this. The order of these priorities has been reversed on page 9.

16. In methods- 30-50 interviews of 40 mins in length, plus observations, seems like a ton of data-- why did you choose those numbers? What if you reach saturation before hand? What is the justification behind it?

It can be challenging to predict the exact the number of interviews needed to gain the necessary information and the point where subsequent interviews would not yield new insights. As a general approach, we are informed by relevant literature and Malterud, Siersma and Guassora's (2016) model for generating "information power" (detailed on page 14).

Study 2.2 "Patients' supportive needs" is a mixed-method study, and as such, the number of interviews to be conducted is also informed by the sample size needed for the quantitative survey, which forms a component of the interview. The following sentence has been added to the "Sample size ranges" section of page 14 to reflect this:

The number of proposed interviews in Study 2.2. is also guided by the sample size needed for the clinician readiness survey that is undertaken as part of the interview.

Reviewer: 2
General comments

1. The authors mention implementation science approaches to these studies. Is there a specific framework being followed?

We thank Reviewer 2 for posing this query and giving us an opportunity to include this detail. It is desirable that what is learned about the structure and organisation of services in this investigation will better position us to select the most appropriate implementation science framework for the subsequent activities of the CRE.

On page 10 the following text has been added to convey this:
This understanding will inform subsequent phases of research undertaken by the Centre and will guide the selection of implementation approaches, some of which will focus on developing and testing tailored interventions to address identified evidence-practice gaps.

Specific comments

2. How will the authors observe ad hoc collaboration between professionals? Will the half day observations be sufficient to observe this?

General observation in the cancer outpatient clinics will provide opportunity to observe processes and practices of care, which would include the ad hoc forms of collaboration that occur. A useful feature of the design is that we have the flexibility to allocate observations to specific cancer outpatient clinics in Stage 2, based on what we learn from Stage 1 observations. This will allow us to better target the observation of ad hoc collaborations and any other interactions of interest. The half-day sessions will take place over a prolonged period and the data collected will be engaged with in conjunction with the interview and multidisciplinary team meeting observation data. Cancer multidisciplinary team meetings also provide opportunity to observe ad hoc collaboration, where for example, a clinician may discuss a patient whose case is not registered on the agenda in advance.

3. What are the PRMs that are going to be introduced?

These details are now included on page 8:

The Edmonton Symptom Assessment Scale²³, Distress Thermometer²⁴ are PRMs which will be introduced into routine cancer care in some of the outpatient clinics (OPCs) during the period of study, likely changing the management of patients' supportive needs.[25-27]

4. Are there any other methods that could be used in combination with observation of clinician practise? Clinicians may be biased in their practice from the knowledge that they are being observed. Further, if no patient interactions are observed, the authors will be relying on clinician self-report? Whilst this may unavoidable, could the data be supported by medical record notes of decision making, electronic referrals etc?

We agree that the observer effect or Hawthorne effect is unavoidable and therefore important to acknowledge and explore during analysis and in the reporting of this study. There is a large literature however, showing that people accommodate to being observed very rapidly and the observer effect is rarely as large as people often believe.[1-3]

We are grateful for Reviewer 2's suggestions for additional methods that could help to triangulate our data. We purposefully allocate time between Stage 1 and Stage 2 of the investigation to consider any amendments to the design or methods of data collection, and we will consider these suggestions at that time.

References

1. Jones, S. Was there a Hawthorne Effect? *Am. J. Sociol.* 1992;98:451-68. doi: 10.1086/230046
2. Kohli, E., Ptak, J., Smith, R., Taylor, E., Talbot, E., & Kirdand, K. Variability in the Hawthorne Effect with regard to hand hygiene performance in high- and low-performing inpatient care units. *Infect. Control Hosp. Epidemiol* 2009;30, 222-225. doi:10.1086/595692
3. Paradis, E, Sutkin, G. Beyond a good story: from Hawthorne Effect to reactivity in health professions education research. *Med Educ*, 2017; 51: 31-39. doi:10.1111/medu.13122

5. Please describe who will conduct the qualitative analysis and their relationship to the research team/authors

These details are now included on page 15.

6. BMJ guidelines state that the dates of the study should be included in the manuscript. Please include.

These details are now included on page 13.

7. The main outcomes of the studies need to be described

The following text has now been added in p 22:

The main outcomes of the investigation are derived from the aims. Through this investigation we will generate a rich and comprehensive description of the organisation and practice of MDC. We will document practices of clinical decision-making including the role of the MDT in such practice. We will produce an account of how patient supportive needs are engaged with and the implications that the introduction of PRMs has for this engagement. The factors which impede or promote practice will be identified and used to inform subsequent CRE-ISO activity.

Formatting Amendments

1. No checklist page number:

Please complete your reporting checklist by indicating the page number where each point can be found.

Please find the Consolidated Criteria for Reporting Qualitative Research Checklist with completed page numbers attached. We have added the details shown below to the manuscript in order to comply with the checklist.

Checklist items 2-4

On page 15 it now reads:

Data will be collected by GA (Senior Research Fellow, Ph.D., male), BNGE (Research Fellow, Ph.D., female) and TW (Research Associate, Masters, female) who are trained in qualitative, quantitative and mixed-methods investigation.

Checklist items 6 & 7

On page 14 the text now reads:

Recruitment: Information about the research including its aims, an introduction to the researchers undertaking data collection, contact details and participant information sheets will be disseminated through email, flyers and posters, and through information sessions for target professional groups at each site.

Checklist item 24.

On page 21 the text now reads:

The discursive data will be coded by two researchers and thematically analysed, oriented by an inductive approach

2. Figure 3 citation missing:

The in text citation for "Figure 3" is missing in your main text of your main document file. Please amend accordingly.

The citation is now included on pages 18 and 20.

VERSION 2 – REVIEW

REVIEWER	Mackenzi Pergolotti ReVital Cancer Rehabilitation Program, Select Medical, USA I receive a salary from Select Medical.
REVIEW RETURNED	03-Aug-2019

GENERAL COMMENTS	Aims: 1- Develop and understanding of cancer services in outpatient clinics by characterizing the organization and practice of MDC 2- Explore key areas- clinical decision making, engagement with patients supportive needs 3- Identify barriers to, facilitators of, the delivery of quality care Concerns: 2- need more explanation into the definition of “engagement with patient supportive needs” how that will be measured specifically and how it will differ from and similar to clinical decision making 3-How is quality of care define here? What is considered good and “poor” quality? Abstract: In abstract- study two, authors mention observations of practice, then later talk about interviews- if doing both it will need to be stated up front. Engagement with patients supportive needs = supportive care provision? By adding surveys – of clinicians- before and after PRM- or the implications of the PRM’s – in terms of how patients are accessing supportive care? I need clarification on the connection between these concepts- PRM, supportive care, engagement with patient supportive needs (clinical team engagement? Patient engagement?) and decision making. How will use of distress scales, do the authors believe will change clinical decision making? In strengths and limitations section authors discuss a 2-stage roll-out which makes me think you are testing the implementation of MCD- is the study to better understanding the services already provided, not to build in services then study the implementation- correct? This needs clarifying throughout manuscript. Introduction: 3rd sentence needs rewording- too confusing the way it ends with “in previous 5 years” either take it out of re-word. 4th sentence- cancer is a condition that requires medical and supportive services- I don’t think you need to hedge that statement as long as you have citations for it. Last sentence- and enable (s?) patients Second paragraph- what type of work was undertaken to translate principles? Are these the principles that are accepted as best practice? Also, authors state there is “substantial evidence..regarding gaps in care” does this mean there is already evidence describing current care? These interconnected elements- are already in current care- or recommended as best evidence? How is decision making similar to patient engagement- Figure one needs more description to be really helpful. Please define supportive care services (or supportive needs—do they lead to supportive services?) Treatment planning would come after evaluation by professional- no?
---

	Need more build up of argument on how use of PRM (especially those chosen) will identify needs and how they are introduced- is this the intervention of study here? Some of the PRM paragraph could be put into methods or limitations section regarding scope of paper. Allied health professional as mentioned as part of core team and again as other expertise? Aim to develop an understanding of current care MDC (plus PRM?) or new MDC Decision-making “studies reveal diversity in organization.. “ This sentence needs clarification- how is diversity defined? These studies just describe the organization of the team meetings or do they state what makes them higher quality is to have diversity etc? Second- if there already is literature describing the teams- what gap does this study fill? Also states they are the fulcrum, and a few sentences later mentions decisions and recommendations are not followed or made in the meetings? Understand what behavior exactly? Not following recommendations- or working meetings? How will understanding this behavior do for the field? What is the gap and question regarding coordination of care? The health economics studies are not included here- so I would delete Why is it important to characterize or describe the current care over and above what has already been written? How will the authors determine the impact/ or describe MCD and clinical decision making while integrating a new intervention PRM-timeline needs clarification. What about barriers and facilitators of PRM? How will the barriers and facilitators of MCD be best understood if it is already current, yet not evidence-based? Methods- State clearly what methods will be harnessed (?) from ethnography or that you are using a method/type of ethnography Philosophical position is helpful, I am still unclear-“underpinned?” so.. there are two positions within one position- ? please clarify Design What is being implemented? Once aims are clarified methods section needs organization as it fits the gaps identified in introduction and argument. Recruitment is concerning, is there evidence behind recruitment by email? Will that be sufficient? Do you already have buy in from these facilities? The amount of interview and observational data is still immense. I need to understand the argument for this before I would say this amount of assessment is needed above and beyond what has already been done and how will this address the gaps? Why is the gender/sex of the authors listed? Is that relevant-if so – how? Are patient supportive needs measured only by PRM? What if they have a need but still do not get care, or want care but are unable to for another reason- how will this data be managed/understood? Modifications by IRB are understood as standard practice and do not need to be spelled out here Main outcomes are derived from the aims- meaning ? please clarify here What other than looking at are the plans for UK patient experience survey and how will that answer your aims? Or the question here?
--	--

REVIEWER	Kristen McCarter University of Newcastle, Australia
REVIEW RETURNED	18-Jul-2019

GENERAL COMMENTS	The authors have addressed all of my concerns and I believe improved the quality of the paper. This paper represents the initial stages of some important work.
---

VERSION 2 – AUTHOR RESPONSE

Reviewer: 1.

Comment 1:

The authors have addressed all of my concerns and I believe improved the quality of the paper. This paper represents the initial stages of some important work.

Response:

We thank the reviewer for their previous feedback which served to strengthen the paper.

Reviewer 2.

Comment 1:

Aims:

1. 1- Develop and understanding of cancer services in outpatient clinics by characterizing the organization and practice of MDC
- 2- Explore key areas- clinical decision making, engagement with patients' supportive needs
- 3- Identify barriers to, facilitators of, the delivery of quality care

Concerns: 2- need more explanation into the definition of "engagement with patient supportive needs" how that will be measured specifically and how it will differ from and similar to clinical decision making

Response 1:

The elements portrayed in Figure 1 play an important role in MDC and, as depicted, overlap. For example, some of the decision-making examined in "Study 2.1. Clinical decision-making" relates to patient supportive needs, while others relate to treatment options relating to cancer type and stage. Equally, "Study 2.2 Patients' supportive needs" examines the current practices for identifying and responding to the supportive needs of patients, some of which will include how these issues impact other clinical decisions.

The current definition on page 7 explains and provides examples of the medical and non-medical patient supportive needs that health professionals engaged with. To further clarify this explanation, we include “anti-cancer treatment with curative intent” to contrast with supportive needs care. This now reads:

In addition to anti-cancer medical treatment with curative intent, people with cancer also need medical (e.g., pain relief) and non-medical supportive care (e.g., information needs). (p.7)

Healthcare professionals’ engagement with supportive needs will be examined through Study 2.2, with the anticipated outcome of producing “an account of how patient supportive needs are engaged with and the implications that the introduction of PRMs has for this engagement” (p.25). In the Methods and Analysis section, Study 2.2 (p.20), provides a full detailing of the methods which will be used to achieve this outcome.

Comment 2:

3-How is quality of care define here? What is considered good and “poor” quality?

In abstract- study two, authors mention observations of practice, then later talk about interviews- if doing both it will need to be stated up front.

Response 2:

In the manuscript we follow the Cancer Australia (the peak Australian cancer organization) definition of the ideal standards of multidisciplinary care:

Multidisciplinary care (MDC) is accepted as a best practice model of care provision for oncology services in Australia, and in advanced health systems globally.⁶⁻⁹ MDC seeks to promote equitable, evidence-based care by a team that combines relevant expertise and enable patients to be involved in decision-making concerning their care.⁶⁻¹⁰ (p.7)

A key strength of the study is that through interviews with oncology health professionals we will gain their perspectives about good or poor care and the contributing factors; deepening our understanding. As noted on page 13:

Barriers and facilitators to the provision of MDC (RA3) will be elicited as part of the interviews in each study and enhanced by our observational studies in Stages One and Two.

We thank the reviewer for this suggestion. We have reviewed the presentation of information about the methods used in Stage 2 studies and feel it is clear that interviews are used to complement the observations:

In Stage Two, observations of practice will continue, to deepen our understanding, and to inform two focused studies. The first will explore decision-making practices and the second will examine how staff engage with patients’ needs; both studies involve interviews, to complement observation. (p.4)

Comment 3:

Engagement with patients supportive needs = supportive care provision?

Response 3:

Engagement with patient supportive needs covers both identifying and responding to patient supportive needs. This is detailed on page 8:

Meeting these needs requires identification, prioritisation, timely access to relevant professional expertise and treatment planning (e.g. in consultation with a psychologist).18-21

Comment 4:

By adding surveys – of clinicians- before and after PRM- or the implications of the PRM's – in terms of how patients are accessing supportive care? I need clarification on the connection between these concepts- PRM, supportive care, engagement with patient supportive needs (clinical team engagement? Patient engagement?) and decision making.

How will use of distress scales, do the authors believe will change clinical decision making?

Response 4:

We are interested in understanding how patients support needs are engaged with and the implications that patient reported measures (PRMs), as a new tool in some settings, will have for identifying and addressing these needs. Learning about the views of health professionals, as end-users of PRMs, is critical to developing this knowledge. Including the clinician readiness survey (specifically designed in relation to PRMs) provides insight into their attitudes, including how they may (or may not) change following the adoption of PRMs. This is described on pages 7, 8, 11 and 20.

As previously detailed: “Study 2.2 Patients’ supportive needs” examines the current practices which engage with the supportive needs of patients (of which decision-making would form only one component) (p.20), whereas Study 2.1, narrows the focus to the processes of decision-making and unlike Study 2.2, will include coverage of anti-cancer medical treatment with curative intent.

We interpret “How will use of distress scales, do the authors believe will change clinical decision making?” as referring to the use of the Distress Thermometer. If so, on page 8, we provide the reader with a specific reference concerning the use of the Distress Thermometer (DT), and subsequently cite research which has piloted and developed PRMs, including the DT in clinical care:

The Edmonton Symptom Assessment Scale²⁵, Distress Thermometer²⁶ are PRMs which some of the outpatient clinics (OPCs) will be introducing into routine cancer care during the period of study, and are expected to change the management of patients’ supportive needs.²⁷⁻²⁹

Watanabe SM, Nekolaichuk CL, Beaumont C. The Edmonton Symptom Assessment System, a proposed tool for distress screening in cancer patients: development and refinement. *Psychooncology* 2012;21:977-85. doi:10.1002/pon.1996

National Comprehensive Cancer Network (NCCN). NCCN Clinical Practice Guidelines in Oncology (NCCN Guidelines®) for distress management 2016.

Cancer Institute NSW. Patient-reported measures (PRMs) 2018. <https://www.cancer.nsw.gov.au/how-we-help/quality-improvement/patient-reported-measures> (accessed 30 August 2018).

Girgis A, Delaney GP, Miller AA. Utilising ehealth to support survivorship care. *Cancer Forum* 2015; 39:86-89.

Girgis A, Durcinoska I, Levesque JV, et al. eHealth system for collecting and utilizing patient reported outcome measures for personalized treatment and care (PROMPT-Care) among cancer patients: mixed methods approach to evaluate feasibility and acceptability. *J Med Internet Res* 2017;19:e330. doi:10.2196/jmir.8360

The introduction of a new tool is a change to existing care, however the specific ways in which this changes practice, (including clinical decision-making), will be explored in Study 2.2.

Comment 5:

In strengths and limitations section authors discuss a 2-stage roll-out which makes me think you are testing the implementation of MCD- is the study to better understanding the services already provided, not to build in services then study the implementation- correct? This needs clarifying throughout manuscript.

Response 5:

Our study is observational; we are not 'testing implementation of MDC', we are observing it in an attempt to understand how it is being implemented in practice.

The two-stage roll-out refers to the design used in the research, and we have revised the wording on page 6 to ensure this is clear:

A two-stage research design facilitates better targeting of observations and interviews in the second stage.

We further clarify this in the Design section. The opening sentence was revised to now read:

A staged design will be used (p.12).

The two-stage design is described in the Design section and each stage and study comprehensively detailed in the Data Collection section (pp.17-20). Further “Table 1. Study design features” (p.16) provides a visual depiction for the reader.

Comment 6:

Introduction

3rd sentence needs rewording- too confusing the way it ends with “in previous 5 years” either take it out of re-word.

Response 6:

We have revised this sentence to read:

Survival rates are improving, with over 400,000 Australians living 5 years post cancer diagnosis. (p.7)

Comment 7:

4th sentence- cancer is a condition that requires medical and supportive services- I don't think you need to hedge that statement as long as you have citations for it.

Response 7:

We have revised this sentence and added a citation. On page 7 it now reads:

Cancer is understood as a condition requiring medical and supportive care services from diagnosis through long-term survivorship.

Comment 8:

Last sentence- and enable(s?) patients

Response 8:

We have replaced “enable” with “enables” (p.7).

Comment 9:

Second paragraph- what type of work was undertaken to translate principles? Are these the principles that are accepted as best practice?

Response 9:

Given the vast scope of these efforts, we now include citation of relevant literature, should the reader wish to explore the history of the development and implementation of MDC in Australia. Following the sentence “Work has been undertaken to translate MDC principles into practice; but significant challenges remain”(p.7), we inserted citations to the following literature:

Zorbas H, Barraclough B, Rainbird K, et al. Multidisciplinary care for women with early breast cancer in the Australian context: what does it mean? *Med J Aust* 2003;179:528-31.

Rankin NM, Lai M, Miller D, et al. Cancer multidisciplinary team meetings in practice: results from a multi-institutional quantitative survey and implications for policy change. *Asia Pac J Clin Oncol* 2018;14:74-83. doi:10.1111/ajco.12765

Wilcoxon H, Luxford K, Saunders C, et al. Multidisciplinary cancer care in Australia: a national audit highlights gaps in care and medico-legal risk for clinicians. *Asia Pac J Clin Oncol* 2011;7:34-40. doi:10.1111/j.1743-7563.2010.01369.x

MDC is accepted as best practice in Australia, and we cited information from Cancer Australia to support this in the manuscript (citation included below).

Multidisciplinary care (MDC) is accepted as a best practice model of care provision for oncology services in Australia, and in advanced health systems globally.6-9 MDC seeks to promote equitable, evidence-based care by a team that combines relevant expertise and enable patients to be involved in decision-making concerning their care.5,10

5. Cancer Australia. All about multidisciplinary care. <https://canceraustralia.gov.au/clinical-best-practice/multidisciplinary-care/all-about-multidisciplinary-care> (accessed 20 July 2018).

Comment 10:

Also, authors state there is “substantial evidence..regarding gaps in care” does this mean there is already evidence describing current care?

Response 10:

The literature concerning gaps is cited in this sentence (p.7). The issues arising in relation to each of the elements is then described and cited in pages 7 through to 11. We draw upon published health district, Australian and international literature to support these arguments. This literature base provides us with a set of issues which we use to guide our studies within the specific services in the two local health districts (LHDs).

The research described here will characterise MDC provision in two metropolitan Local Health Districts (LHDs), government agencies responsible for managing and providing public health services within a specified geography, usually of approximately one million residents in order to provide a foundational understanding of the realities of oncology service provision. This understanding will inform subsequent phases of research undertaken by the Centre and will guide the selection of implementation approaches, some of which will focus on developing and testing tailored interventions to address identified evidence-practice gaps. (p.10)

We select qualitative methods that will allow us to explore these issues while taking account of the nuances and site-specificities (supported by the “profiles and structural descriptions of each hospital setting” (p.12)). A contextualised understanding is critical to the development of interventions sufficiently tailored to sites (noted above). This will also inform our assessment of the applicability of these findings to services outside of our study settings.

Comment 11:

These interconnected elements are already in current care- or recommended as best evidence? How is decision making similar to patient engagement- Figure one needs more description to be really helpful?

Response 11:

Figure 1 presents elements of MDC care which were identified through “review of research literature, national health sector reports and local cancer plans” (p.7). These elements correspond with the MDC approach to best practice and are currently part of care. This figure provides a visual overview to foreground the detailed description of each which follows on pages 7-9.

As the question about the relationship between patients’ needs and decision-making is raised previously, we provide this response again here (note that we interpret the reviewer’s term ‘patient engagement’ as being shorthand for engagement with patient supportive needs).

The elements portrayed in Figure 1 play an important role in MDC and, as depicted, overlap. For example, some of the decision-making examined in “Study 2.1. Clinical decision-making” will be relate to patient supportive needs. However, the two focused studies provide us an opportunity to focus more deeply with each of these specific elements of care.

Comment 12:

Please define supportive care services (or supportive needs—do they lead to supportive services?)

Response 12:

As per our response to an earlier comment, the current definition on page 7 explains and provides examples of the medical and non-medical patient supportive needs that health professionals engage with. To further clarify this explanation, we use the term “anti-cancer treatment with curative intent” as a means of contrasting with supportive needs care provision. This now reads:

In addition to anti-cancer medical treatment with curative intent, people with cancer also need medical (e.g., pain relief) and non-medical supportive care (e.g., information needs). (p.7)

In this same paragraph we then specify what this would involve on the part of services. Here, we included an example to aid understanding. The text on pages 7 and 8 was revised to read:

Meeting these needs requires identification, prioritisation, timely access to relevant professional expertise and treatment planning (e.g. in consultation with a psychologist). (p.8)

Comment 13:

Treatment planning would come after evaluation by professional- no?

Response 13:

We have rearranged the order. This sentence now reads:

Meeting these needs requires identification, prioritisation, timely access to relevant professional expertise and treatment planning (e.g. in consultation with a psychologist). (p.8)

Comment 14:

Need more build up of argument on how use of PRM (especially those chosen) will identify needs and how they are introduced- is this the intervention of study here?

Response 14:

The study described in the protocol is observational, not interventional. PRMs are part of a state-wide roll out mandated by government authorities; the timing of this roll out coincidentally allows us to observe the implications for engagement with patients supportive needs. As stated on page 8:

PRMs which will be introduced into routine cancer care during the period of study, likely changing the management of patients' supportive needs.

To remove ambiguity, we have revised this to read:

PRMs which some of the outpatient clinics (OPCs) will be introducing into routine cancer care during the period of study, and are expected to change the management of patients' supportive needs. (p.8)

Comment 15:

Some of the PRM paragraph could be put into methods or limitations section regarding scope of paper.

Response 15:

See response to previous point. No changes made to manuscript.

Comment 16:

Allied health professional as mentioned as part of core team and again as other expertise?

Response 16:

There is a wide range of professional groups involved in allied health; a specific MDT may involve some allied health disciplines, but not all. This reflects the current approach to MDT membership whereby it is recognised that patient needs may change over the course of care and the professional expertise in the care team may change to better address these needs. Citations are provided to the relevant, supporting literature.

Comment 17:

Aim to develop an understanding of current care MDC (plus PRM?) or new MDC

Response 17:

The study aims to investigate current practice and the implications of PRMs, as they are introduced. No changes made to manuscript.

Comment 18:

Decision-making "studies reveal diversity in organization.. " This sentence needs clarification- how is diversity defined?

Response 18:

“Diversity” is not used here as a technical term; it simply refers to variability. The remainder of this sentence specifies the key areas of this variation. We have replace “diversity” with “variability”:

Studies reveal variability in the organisation of MDTMs, who is discussed and when, and the ways that clinicians engage to support clinical decision-making. (p.9)

Comment 19:

These studies just describe the organization of the team meetings or do they state what makes them higher quality is to have diversity etc? Second- if there already is literature describing the teams- what gap does this study fill?

Response 19:

Observation of MDTMs, as a standalone method, is not intended to address the issues identified in the literature; the studies, which use a suite of methods, are designed to contribute to knowledge in the key areas identified. We have reviewed the protocol document and do not believe this is intimated at any point. The data from observations is considered in relation to interview and general observational data to understand clinical decision-making practices, within and outside of MDTMs and the role that these meetings play in supporting decision-making.

Comment 20:

Also states they are the fulcrum, and a few sentences later mentions decisions and recommendations are not followed or made in the meetings?

Understand what behavior exactly? Not following recommendations- or working meetings? How will understanding this behavior do for the field?

Response 20:

It is the case that MDTMs play a critical role in decision-making, and also the case that recommendations reached are not always implemented. As noted:

While MDTMs play a central role, clinicians make decisions outside MDTMs, without any consultation during routine care provision, or in consultation with other clinicians who may or may not be part of the team (p.9).

The research will help us to understand decision-making behaviour within and outside of MDTMs. For example, previous research indicates differences in:

who is discussed and when, and the ways that clinicians engage to support clinical decision-making; in-depth qualitative research helps us to explore and understand decision-making behaviour. (p.9).

This has implications for patients and their care, as well as fellow health professionals involved in care provision. By understanding how decisions are actually made in practice in the targeted sites, we will be better placed, in any future study, to design an intervention that seeks to change the way decisions are made.

Comment 21:

What is the gap and question regarding coordination of care?

Response 21:

Reviews of the relevant plans and literature highlight the important role of coordination in MDC care and as described the challenging task of ensuring continuity of care within the existing service context (p.9). We do not pose a question, but consider the issues raised in this section as we undertake the individual studies described.

Comment 22:

The health economics studies are not included here- so I would delete

Response 22:

References to health economics studies have been removed on pages 10, 11 and 25.

Comment 23:

Why is it important to characterize or describe the current care over and above what has already been written?

Response 23:

To our knowledge, care has not been characterised in this way in the LHDs, and we look forward to making this contribution. The study described in the protocol is the first in a series looking to advance implementation science in oncology; identifying general barriers and facilitators of the provision of evidence-based care in the targeted settings is one of our starting points.

Comment 24:

How will the authors determine the impact/ or describe MCD and clinical decision making while integrating a new intervention PRM- timeline needs clarification.

Response 24:

“Table 1. Study design features” (p.16) shows the order of the studies; Study 1. Multi-site characterisation study (exploring MDC), takes place in Stage 1, while in Stage 2, which follows, the two focused studies will be take place. As depicted in Table 1, Study 2.1 and Study 2.2. will be largely conducted in separate LHDs. The section that follows, goes on describe the studies undertaken at each stage in detail (pp.17-21). This includes description of the data collection and analysis. As stated, the CRE is not integrating or implementing a PRM intervention.

Comment 25:

What about barriers and facilitators of PRM?

Response 25:

The barriers and facilitators to PRMs will be examined through the interviews:

Topics will include: models of care; their individual role; identification and screening; management; and any barriers and facilitators experienced in identifying and responding to supportive needs. The interviews will also explore preparations for and expectations of PRMs. (p.20).

Comment 26:

How will the barriers and facilitators of MCD be best understood if it is already current, yet not evidence-based?

Response 26:

The literature would suggest there are difficulties translating the principles of MDC into practice and that there are “gaps between optimal evidence-based care and current care” (p.7). We have added “In this regard” to ensure the link between sentences is clear.

The text on page 7 now reads:

Work has been undertaken to translate MDC principles into practice; but significant challenges remain. In this regard, substantial evidence points to gaps between optimal evidence-based care and current care.13-17

The barriers and facilitators to the provision of MDC “will be elicited as part of the interviews in each study and enhanced by our observational studies in Stages One and Two” (p.13). The Methods and Analysis section subsequently details the specific processes involved.

Comment 27:

Methods- State clearly what methods will be harnessed (?) from ethnography or that you are using a method/type of ethnography

Response 27:

The type of ethnographic approach is detailed in the first paragraph of the Methods and Analysis section on page 12:

Methods adapted from ethnography will be harnessed, to facilitate intensive and efficient data collection; rapid ethnography will be used to reduce the burden of research on organisations and participants while maintaining rigour and richness.

The specific methods are detailed in the next paragraph:

The methodological principles embedded in the design are: exploring everyday practices and the organisational relations that shape them, by observing participants at work in OPC settings; harnessing a plurality of participant perspectives, by interviewing professionals about their work; and generating rich descriptive accounts of MDC contexts by using analytical and interpretative approaches formulated for multi-site data. (p.12).

We have reviewed this content, and believe it reads clearly. No changes made to manuscript.

Comment 28:

Philosophical position is helpful, I am still unclear-“underpinned?” so.. there are two positions within one position- ? please clarify

Response 28:

The critical realist philosophical position is taken, which is made up of “a realist ontological assumption and a critical epistemology”. This information is provided to ensure a clear distinction from a “direct realist” position, which would not contain a critical epistemological orientation. For ease, we replace the term “underpinned” with “comprised of” on page 12.

Comment 29:

Design

What is being implemented?

Response 29:

We have revised the wording of this sentence to read “A staged design will be used” (p.12), removing the word implemented, to avoid confusion.

Comment 30:

Once aims are clarified methods section needs organization as it fits the gaps identified in introduction and argument

Response 30:

We appreciate that the design is complex, with individual activities addressing or informing multiple aims, but after reflection we have concluded that the current structure is best suited. As it stands, we address each of the areas required by the journal guidelines, the relevant components of the COREQ and the specific design in a logical and coherent fashion. The first paragraph of the Design section explicitly states how each of the research aims are addressed in the research design.

Comment 31:

Recruitment is concerning, is there evidence behind recruitment by email? Will that be sufficient? Do you already have buy in from these facilities?

Response 31:

Recruitment via email is a common strategy for studies in this area (e.g. Rankin et al., 2018) and is used in conjunction with other methods, for example:

Information about the research including its aims, an introduction to the researchers undertaking data collection, contact details and participant information sheets will be disseminated through email, flyers and posters, and through information sessions for target professional groups at each site. (p.14)

Further we employ snowballing and direct requests to MDT Chairs. (pp.14-15). We use a range of recruitment options to minimise the burden on the health professionals approached.

In terms of “buy in”, the CRE is very fortunate to count among its leaders key figures in the LHDs, who guide this work. To clarify this connection, we revise text in the “Research Team” section to read:

Fieldwork will be guided by principal investigators with direct clinical and research experience in oncology service provision, in leadership roles in the LHDs. (p.15)

Further, as noted above, we provide information sessions for target professional groups at each site, which facilitates “buy-in” from staff at the services.

Comment 32:

The amount of interview and observational data is still immense. I need to understand the argument for this before I would say this amount of assessment is needed above and beyond what has already been done and how will this address the gaps?

Response 32:

In the first review, one reviewer advised:

“In methods- 30-50 interviews of 40 mins in length, plus observations, seems like a ton of data--why did you choose those numbers? What if you reach saturation before hand? What is the justification behind it?”

To address this concern we provided a justification for the sample sizes and revised the text accordingly.

“It can be challenging to predict the exact the number of interviews needed to gain the necessary information and the point where subsequent interviews would not yield new insights. As a general approach we are informed by relevant literature and Malterud, Siersma and Guassora’s (2016) model for generating “information power” (detailed on page 14).

Study 2.2 “Patients’ supportive needs” is a mixed-method study, and as such, the number of interviews to be conducted is also informed by the sample size needed for the quantitative survey, which forms a component of the interview. The following sentence has been added to the “Sample size ranges” section of page 14 to reflect this:

The number of proposed interviews in Study 2.2. is also guided by the sample size needed for the clinician readiness survey that is undertaken as part of the interview.”

Our earlier response is not referenced in this new comment, and without guidance about why our response was not satisfactory, it is difficult to address the specific concern.

It might be helpful to note that in addition to detailing the process and basis for determining sample sizes, we also provide evidence from the current published literature. In this same paragraph, we present citations for relevant studies which use comparable methods, in order to highlight that these sample size ranges are consistent with the published literature and expected:

The number of sessions proposed for the observations of the OPCs and MDTMs are comparable with similar qualitative studies⁵⁵ and are appropriate for multi-site ethnographic studies.⁵⁶ (p.15).

The addressing of gaps in understanding is not covered in the Sample Size Range section, rather we outline the key issues arising in the literature (pp.7-9), formulate our aims (p.11), and describe how these will be addressed throughout the Methods and Analysis section (pp.12-21), present the Main Outcomes (p.22), and articulate the contribution this will make to the field in the Impact and Significance section (p.21).

Comment 33:

Why is the gender/sex of the authors listed? Is that relevant-if so – how?

Response 33:

These details have been included to adhere to the consolidated criteria for reporting qualitative research (COREQ); a well-established qualitative reporting guideline nominated by BMJ Open. Citation is provided for the paper which provides the full details and justification for each of the checklist items.

Tong A, Sainsbury P, Craig J. Consolidated criteria for reporting qualitative research (COREQ): a 32-item checklist for interviews and focus groups. *Int J Qual Health Care* 2007;19:349-57.
doi:10.1093/intqhc/mzm042

Comment 34:

Are patient supportive needs measured only by PRM? What if they have a need but still do not get care, or want care but are unable to for another reason- how will this data be managed/understood?

Response 34:

In current practice, at the site where PRMs are planned for introduction, we understand that patients' supportive needs are identified as part of clinical interactions. PRMs offer an approach and specific set of tools which likely change "the management of patients' supportive needs" (p.8). We design Study 2.2. to understand current practices of engaging with patient supportive needs and the implications of PRMs for these practices. The selected methods (p.20) allow us to explore other measurements or approaches that exist. The interviews with staff members will help us to understand any shortcomings (such as those mentioned in the comment). The plan for the management and analysis of this data can be found on pages 21-22.

While we agree that it would be useful to understand patients' views about their supportive needs and how well they were or were not met, this is not the purpose of our study.

Comment 35:

Modifications by IRB are understood as standard practice and do not need to be spelled out here

Response 35:

We agree and have deleted this sentence from page 18.

Comment 36:

Main outcomes are derived from the aims- meaning ? please clarify here

Response 36:

The main outcomes are formed on the basis of the aims, to ensure coherence between aims, methods and anticipated outcomes. For ease of understanding we have revised “derived from” to read “aligned with” on page 22.

Comment 37:

What other than looking at are the plans for UK patient experience survey and how will that answer your aims? Or the question here?

Response 37:

Analysis of the UK Patient Experience Surveys is undertaken as part of separate patient-focused studies within the CRE. As per your request in the first review to “take room to really describe why patients were not included”, we responded:

“We are in complete agreement with reviewer 1 that the experiences of patients are vital and believe that we will better engage with patient perspectives through dedicated patient-centred studies within the Centre for Research Excellence in Implementation Science in Oncology (CRE-ISO) program of activity. The text in section “Patient and Public Involvement Statement” (p 24) has been elaborated to emphasise this point and to note the importance of learning about the facilitators and barriers patients experience.

The text now reads:

This project will help us to describe service structure and organisation, and to learn from the perspectives of MDC providers. The experiences of patients are vital to understanding these services, including their views on the barriers and facilitators to high-quality care. Patient experiences will be engaged with in dedicated, patient-centred studies within the CRE-ISO program of research. For example, in preparation we have already looked at data from UK Patient Experience Surveys⁶³ and are in the process of accessing Australian data from a similar source”.

As noted, our research on the UK Patient Experience Surveys is provided to illustrate our commitment to undertaking patient-centred studies, within the larger program of the CRE, in addition to the services-focused research described in this protocol.